# MAGICTAILOR:
# COMPONENT-CONTROLLABLE PERSONALIZATION IN TEXT-TO-IMAGE DIFFUSION MODELS

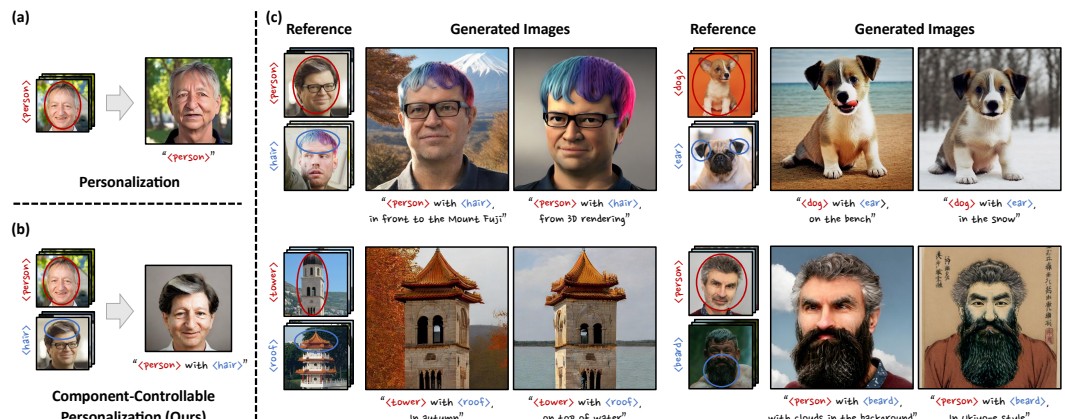

Figure 1: **(a) Illustration of personalization,** demonstrating how text-to-image (T2I) diffusion models can learn and reproduce a visual concept from given references. **(b) Illustration of component-controllable personalization,** depicting a newly formulated task that aims to modify a specific component of a visual concept during the personalization process. **(c) Example images generated by the proposed MagicTailor,** showcasing the effectiveness of MagicTailor, a novel framework that adapts T2I diffusion models for component-controllable personalization, enabling the generation of text-aligned, visually-coherent, and high-quality images. For clarity, the red and blue circles are used to highlight the target concept and component, respectively.

## ABSTRACT

Recent advancements in text-to-image (T2I) diffusion models have enabled the creation of high-quality images from text prompts, but they still struggle to generate images with precise control over specific visual concepts. Existing approaches can replicate a given concept by learning from reference images, yet they lack the flexibility for fine-grained customization of the individual component within the concept. In this paper, we introduce **component-controllable personalization**, a novel task that pushes the boundaries of T2I models by allowing users to reconfigure and personalize specific components of concepts. This task is particularly challenging due to two primary obstacles: *semantic pollution*, where unwanted visual elements corrupt the personalized concept, and *semantic imbalance*, which causes disproportionate learning of visual semantics. To overcome these challenges, we design **MagicTailor**, an innovative framework that leverages *Dynamic Masked Degradation (DM-Deg)* to dynamically perturb undesired visual semantics and *Dual-Stream Balancing (DS-Bal)* to establish a balanced learning paradigm for visual semantics. Extensive comparisons, ablations, and analyses demonstrate that MagicTailor not only excels in this challenging task but also holds significant promise for practical applications, paving the way for more nuanced and creative image generation. Our code will be released.

# 1 INTRODUCTION

Text-to-image (T2I) diffusion models (Rombach et al., 2022; Saharia et al., 2022; Ramesh et al., 2022; Chen et al., 2023) demonstrate remarkable capabilities in generating high-quality visual content from textual descriptions. These models are able to create images that closely match the provided prompts. But when certain visual concepts are difficult to articulate through natural language, they may face difficulties in accurately incorporating such elements into the generated images. To address this limitation, some approaches (Gal et al., 2022; Ruiz et al., 2023) enable T2I models to learn specific concepts from a few reference images, thus allowing for more faithful integration of these concepts into the generated images. This process, illustrated in Figure 1(a), is referred as personalization. However, existing personalization methods are limited to replicating predefined concepts and lack the capability to offer flexible, fine-grained control over these elements. This constraint significantly limits their practical applicability in real-world scenarios. A concept often comprises multiple components, such as a house consisting of walls, windows, and doors. Therefore, a more sophisticated challenge in personalization is determining *how to effectively control and manipulate these individual components during the personalization process*.

This paper introduces a new task, **component-controllable personalization**, which aims to reconfigure the elements of a personalized concept using additional visual references (see Figure 1(b)). In this task, a T2I model is fine-tuned with reference images and corresponding category labels, allowing it to learn and generate the desired concept along with its specified component. Achieving this capability would not only enable users to refine and customize concepts with precise control but also foster innovation and creativity, paving the way for novel ideas, inventions, and artworks across various creative domains.

A straightforward approach to this task is to treat each component as a separate concept and use existing personalization methods to combine multiple concepts with suitable text prompts. However, this naive strategy falls short in component-controllable personalization due to the inherent complexity of handling visual semantics during learning. One of the key challenges in this task is **semantic pollution** (see Figure 2(a)), where undesired visual semantics inadvertently appear in generated images, thereby *polluting* the personalized concept. This occurs because the T2I model tends to blend visual semantics from different regions during the learning process. Simply masking out unwanted visual elements in reference images is not a viable solution, as it disrupts the overall visual context and leads to unintended compositions. Another significant challenge is **semantic imbalance** (see Figure 2(b)), which causes the T2I model to focus disproportionately on certain aspects, resulting in unfaithful personalization. This issue arises from the semantic disparity between the concept and its components, highlighting the need for an effective learning paradigm to better manage concept-level (*e.g.*, person) and component-level (*e.g.*, hair) visual semantics.

To address these challenges, we present **MagicTailor**, a novel framework that enables component-controllable personalization for T2I models (see Figure 1(c)). As shown in Figure 3, we first employ a text-guided image segmenter to generate segmentation masks for both the concept and its components. Then, we introduce a technique called **Dynamic Masked Degradation (DM-Deg)**, which transforms the original reference images into randomly degraded versions, dynamically perturbing undesired visual semantics. This approach helps suppress the model's sensitivity to irrelevant visual details while preserving the overall visual context, thereby effectively mitigating semantic pollution. Next, we initiate a warm-up phase for the T2I model by jointly training it on these degraded images, using a masked diffusion loss to focus on the desired visual semantics and an attention loss to strengthen the correlation between these semantics and pseudo-words. To tackle the issue of semantic imbalance, we employ **Dual-Stream Balancing (DS-Bal)**, a dual-stream learning paradigm designed for balancing the learning of visual semantics, to launch the second phase. In this paradigm, the online denoising U-Net performs sample-wise min-max optimization, while the momentum denoising U-Net applies selective preserving regularization. This balanced approach ensures more faithful and accurate personalization of the target concept and component.

We validate the effectiveness of MagicTailor through comprehensive qualitative and quantitative experiments, demonstrating that it can achieve state-of-the-art (SOTA) performance in component-controllable personalization. Detailed ablation studies further confirm the impact of the key techniques integrated into MagicTailor. Additionally, we showcase its potential to enable a variety of further applications. In summary, the main contributions of this work are as follows:

Figure 2: **Major challenges in component-controllable personalization.** **(a) Semantic pollution:** (i) Undesired visual elements may inadvertently disturb the personalized concept. (ii) A simple mask-out strategy is ineffective and causes unintended compositions, whereas (iii) our DM-Deg effectively suppresses unwanted visual semantics, preventing such pollution. **(b) Semantic imbalance:** (i) Simultaneously learning the concept and component can lead to imbalance, resulting in unfaithful personalization or component disappearance (here we present a case for the former). (ii) Our DS-Bal ensures balanced learning, enhancing personalization performance.

- We introduce a new task called *component-controllable personalization* for T2I models, enabling precise control over the individual components of concepts during personalization. Moreover, *semantic pollution* and *semantic imbalance* are identified as key challenges in this task.
- We propose *MagicTailor*, a novel framework specifically designed for component-controllable personalization. This framework incorporates *Dynamic Masked Degradation (DM-Deg)* to dynamically perturb undesired visual semantics, and *Dual-Stream Balancing (DS-Bal)* to ensure balanced learning of visual semantics.
- Comprehensive comparisons demonstrate that MagicTailor achieves superior performance in this task. Additionally, ablation studies and further applications highlight the effectiveness and versatility of the proposed method.

## 2 RELATED WORKS

**Text-to-Image Generation.** Text-to-image (T2I) generation has made remarkable advancements in recent years, enabling the synthesis of vivid and diverse imagery based on textual descriptions. Early methods employed Generative Adversarial Networks (GANs) (Reed et al., 2016; Xu et al., 2018; Qiao et al., 2019; Zhu et al., 2019), and auto-regressive transformers (Ding et al., 2021; Ramesh et al., 2021; Ding et al., 2022; Yu et al., 2022) began to show the potential in conditioned image generation. More recently, the advent of diffusion models has ushered in a new era in T2I generation (Li et al., 2024; Saharia et al., 2022; Ramesh et al., 2022; Chen et al., 2023; Xue et al., 2024). Leveraging these models, a range of related applications has rapidly emerged, including image editing (Li et al., 2024; Mou et al., 2024; Huang et al., 2024), image completion and translation (Xie et al., 2023b;a; Lin et al., 2024), and controllable generation (Zhang et al., 2023; Wang et al., 2024b; Zheng et al., 2023). Despite advancements in T2I diffusion models, generating images that accurately capture specific, user-defined concepts remains challenging. This study explores component-controllable personalization, enabling precise adjustment of concept's components through visual references.

**Personalization.** Personalization seeks to adapt T2I models to generate specific concepts using reference images. Initial approaches such as textual inversion (Gal et al., 2022) and DreamBooth (Ruiz et al., 2023) addressed this task by either optimizing a text embedding or fine-tuning the entire T2I model. Additionally, low-rank adaptation (LoRA) (Hu et al., 2021) has been widely adopted by the research community for personalization (Ryu, 2022), offering an efficient and lightweight solution. The scope of personalization has further expanded to accommodate multiple concepts (Kumari et al., 2023; Avrahami et al., 2023; Liu et al., 2023; Gu et al., 2024; Han et al., 2023; Gu et al., 2024). Besides, a growing body of works has explored tuning-free approaches to personalization (Xiao et al., 2023; Li et al., 2023; Shi et al., 2023; Wei et al., 2023; Wang et al., 2024a). However, these methods often rely on training an encoder with extensive domain-specific image datasets. There is also a category of works studying training-free schemes (Jeong et al., 2024; Zhang et al., 2024), but they generally suffer from inferior performance and tortuous inference processes. In light of that, Our MagicTailor goes with a widely-adopted paradigm of test-time optimization to achieve stable performance and precise control over the concept during personalization.

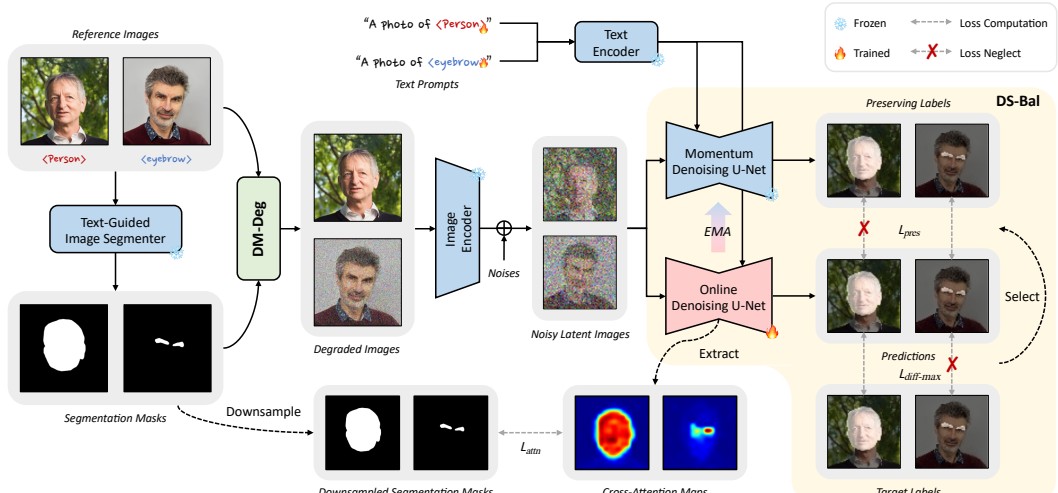

Figure 3: **Pipeline Overview of MagicTailor.** Using reference images as the inputs, MagicTailor fine-tunes a T2I diffusion model to learn both the target concept and component, enabling the generation of images that seamlessly integrate the component into the concept. Two key techniques, Dynamic Masked Degradation (DM-Deg, see Section 3.2) and Dual-Stream Balancing (DS-Bal, see Section 3.3), address the challenges of semantic pollution and semantic imbalance, respectively. For clarity, only one image per concept/component is presented and the warm-up stage is not depicted.

## 3 METHODOLOGY

Let $\mathcal{I} = \{(\{I_{nk}\}_{k=1}^K, c_n)\}_{n=1}^N$ represents a concept-component pair with $N$ samples of concepts and components, where each sample contains $K$ reference images $\{I_{nk}\}_{k=1}^K$ with a category label $c_n$. In this work, we focus on handling one concept and one component to make the task setting more practical. Specifically, we set $N = 2$ and define the first sample as a concept (*e.g.*, dog) while the second one as a component (*e.g.*, ear). In addition, these samples are associated with the pseudo-words $\mathcal{P} = \{p_n\}_{n=1}^N$ serving as their text identifiers. The objective of *component-controllable personalization* is to fine-tune a text-to-image (T2I) model to accurately learn both the concept and component from $\mathcal{I}$. Using text prompts with $\mathcal{P}$, the fine-tuned model is expected to generate images that contain the personalized concept integrated with specified components.

In this section, we begin by providing an overview of the MagicTailor pipeline (refer to Section 3.1). Following this, we delve into its two core techniques: Dynamic Masked Degradation (DM-Deg, see Section 3.2) and Dual-Stream Balancing (DS-Bal, see Section 3.3).

### 3.1 OVERALL PIPELINE

The overall pipeline of MagicTailor is illustrated in Figure 3. The process begins with identifying the desired concept or component within each reference image $I_{nk}$, employing an off-the-shelf text-guided image segmenter to generate a segmentation mask $M_{nk}$ based on $I_{nk}$ and its associated category label $c_n$. Conditioned on $M_{nk}$, we introduce *Dynamic Masked Degradation (DM-Deg)* to perturb undesired visual semantics within $I_{nk}$, addressing *semantic pollution*. At each training step, DM-Deg transforms $I_{nk}$ into a randomly degraded image $\hat{I}_{nk}$, with the degradation intensity being dynamically regulated. Subsequently, these degraded images, along with structured text prompts, are used to fine-tune a T2I diffusion model to facilitate concept and component learning. The model is formally expressed as $\{\epsilon_\theta, \tau_\theta, \mathcal{E}, \mathcal{D}\}$, where $\epsilon_\theta$ represents the denoising U-Net, $\tau_\theta$ is the text encoder, and $\mathcal{E}$ and $\mathcal{D}$ denote the image encoder and decoder, respectively. To promote the learning of the desired visual semantics, we employ the masked diffusion loss, which is defined as

$$\mathcal{L}_{\text{diff}} = \mathbb{E}_{n,k,\epsilon,t}\left[\left\|\epsilon \odot M'_{nk} - \epsilon_\theta(z_{nk}^{(t)}, t, e_n) \odot M'_{nk}\right\|_2^2\right], \quad (1)$$

where $\epsilon \sim \mathcal{N}(0, 1)$ is the unscaled noise, $z_{nk}^{(t)}$ is the noisy latent image of $\hat{I}_{nk}$ with a random time step $t$, $e_n$ is the text embedding of the corresponding text prompt, and $M'_{nk}$ is downsampled from

$M_{nk}$ to match the shape of $\epsilon$ and $z_{nk}$. Additionally, we also incorporate the cross-attention loss to strengthen the correlation between desired visual semantics and their corresponding pseudo-words, formulated as

$$\mathcal{L}_{\text{attn}} = \mathbb{E}_{n,k,t}\left[\left\|A_\theta(p_n, z_{nk}^{(t)}) - M_{nk}''\right\|_2^2\right], \tag{2}$$

when $A_\theta(p_n, z_{nk}^{(t)})$ is the cross-attention maps between the pseudo-word $p_n$ and the noisy latent image $z_{nk}^{(t)}$ and $M_{nk}''$ is downsampled from $M_{nk}$ to match the shape of $A_\theta(p_n, z_{nk}^{(t)})$. Using $\mathcal{L}_{\text{diff}}$ and $\mathcal{L}_{\text{attn}}$, we first warm up the T2I model by jointly learning all samples, aiming to preliminarily inject the knowledge of visual semantics into it. The loss of the warm-up stage is defined as

$$\mathcal{L}_{\text{warm-up}} = \mathcal{L}_{\text{diff}} + \lambda_{\text{attn}}\mathcal{L}_{\text{attn}}, \tag{3}$$

where $\lambda_{\text{attn}} = 0.01$ is the loss weight for $\mathcal{L}_{\text{attn}}$. For efficient fine-tuning, we only train the denoising U-Net $\epsilon_\theta$ in a low-rank adaptation (LoRA) (Hu et al., 2021) manner and the text embedding of the pseudo-words $\mathcal{P}$, keeping the others frozen. Thereafter, we employ *Dual-Stream Balancing (DS-Bal)* to establish a dual-stream learning paradigm to address the challenge called *semantic imbalance*. In this paradigm, the online denoising U-Net $\epsilon_\theta$ conducts sample-wise min-max optimization for the hardest-to-learn sample, and meanwhile the momentum denoising U-Net $\tilde{\epsilon}_\theta$ applies selective preserving regularization for the other sample.

## 3.2 Dynamic Masked Degradation

In this task, one of the major challenges is *semantic pollution*, where undesired visual semantics could be perceived by the T2I model and thus "pollute" the personalized concept. As shown in Figure 2(a.i), the target concept (*i.e.*, person) could be severely disturbed by the owner of the target component (*i.e.*, eye), resulting in a hybrid person. Unfortunately, directly masking out the regions other than the target concept and component would damage the overall visual context, thus leading to overfitting and weird compositions in Figure 2(a.ii). In light of that, the undesired visual semantics of reference images should be processed properly. Hence, we propose *Dynamic Masked Degradation (DM-Deg)* to dynamically perturb undesired visual semantics (see Figure 3), aiming at suppressing the T2I model's perception for them while maintaining overall visual contexts (see Figure 2(a.iii)).

**Degradation Imposition.** In each training step, DM-Deg imposes degradation in the regions outside segmentation masks for each reference image. There are various types of degradation that can be adopted to perturb the visual semantics of an image, such as noise, blur, and geometric distortions, but not all of them are easy to use and compatible with mask operations. In DM-Deg, we choose to employ Gaussian noise due to its simplicity. For a reference image $I_{nk}$, we randomly sample a Gaussian noise matrix $G_{nk} \sim \mathcal{N}(0, 1)$ with the same shape as $I_{nk}$. Then, with the corresponding segmentation mask $M_{nk}$, the imposition of degradation is conducted as

$$\hat{I}_{nk} = \alpha_d G_{nk} \odot (1 - M_{nk}) + I_{nk}, \tag{4}$$

where $\odot$ indicates element-wise multiplication and $\alpha_d \in [0, 1]$ is a dynamic weight used to regulate the degradation intensity for $I_{nk}$. In this way, we can obtain a randomly degraded image $\hat{I}_{nk}$ where the original visual contexts are generally retained. Encountering with $\hat{I}_{nk}$, it is more difficult for the T2I model to fit undesired visual semantics in out-of-mask regions, since these semantics would be randomly perturbed with Gaussian noise at each training step.

**Dynamic Intensity.** Unfortunately, the T2I model may gradually memorize the introduced noise in some cases, especially in the later training, thus leading to noise appearing in generated images. Thus, we design a descending scheme to dynamically regulate the intensity of the imposed noise during training. This scheme adopts an exponential curve that maintains a relatively large intensity in the early and decreases dramatically in the later. Let $d$ denote the current training step and $D$ denote the total training step. The curve of dynamic intensity is defined as

$$\alpha_d = \alpha_{\text{init}}(1 - (\frac{d}{D})^\gamma), \tag{5}$$

where $\alpha_{\text{init}}$ is the initial value of $\alpha_d$ and $\gamma$ is a factor to regulate the descent rate. We empirically set $\alpha_{\text{init}} = 0.5$ and $\gamma = 32$ tuned within the powers of 2. Using such a scheme of dynamic intensity, we can effectively prevent semantic pollution and meanwhile alleviate the memorization of the introduced noise, achieving better generation performance.

## 3.3 DUAL-STREAM BALANCING

Another primary challenge in this task is *semantic imbalance*, which arises from the inherent visual semantic disparity between the target concept and component. Generally, a concept has richer visual semantics than a component (*e.g.* person vs. hair). However, the semantic richness of a component might be greater than a concept in some cases (*e.g.*, simple tower vs. intricate roof). This imbalance complicates the joint learning process which could overemphasis on the concept and component, leading to uncoherent generation of the target concept and even the disappearance of the target component (see Figure 2(b.i)). To address this challenge, we design *Dual-Stream Balancing (DS-Bal)*, which uses online and momentum denoising U-Nets (see Figure 3) to balance visual semantic learning between the concept and component, improving personalization fidelity (see Figure 2(b.ii)).

**Sample-wise Min-Max Optimization.** From a loss perspective, the visual semantics of the concept and component are learned by optimizing the masked diffusion loss $\mathcal{L}_{\text{diff}}$ for all the samples. Unfortunately, this indiscriminate optimization does not allocate adequate learning efforts to the sample that is more challenging to learn, gradually leading to an imbalanced learning process. To overcome this issue, DS-Bal utilizes the online denoising U-Net to learn only the visual semantics of the hardest-to-learn sample at each training step. Inheriting the weights of the original denoising U-Net warmed up by joint learning, here the online denoising U-Net $\epsilon_\theta$ only optimizes the maximum masked diffusion loss among $N$ samples, which is defined as

$$\mathcal{L}_{\text{diff-max}} = \max_n \mathbb{E}_{k,\epsilon,t}\left[\left\|\epsilon \odot M'_{nk} - \epsilon_\theta(z^{(t)}_{nk}, t, e_n) \odot M'_{nk}\right\|^2_2\right], \tag{6}$$

where minimizing $\mathcal{L}_{\text{diff-max}}$ can be considered as a form of min-max optimization (Razaviyayn et al., 2020). The learning objective of $\epsilon_\theta$ may switch across different training steps and is not consistently dominated by the concept or component. Such an optimization scheme can effectively modulate the learning dynamics of different samples and avoid the overemphasis on any particular one.

**Selective Preserving Regularization.** At a training step, the samples neglected in $\mathcal{L}_{\text{diff-max}}$ may suffer from knowledge forgetting. This is because the optimization of $\mathcal{L}_{\text{diff-max}}$, which aims to enhance the knowledge of a specific sample, could inadvertently overshadow the knowledge of the others. In light of this, DS-Bal meanwhile exploits the momentum denoising U-Net to preserve the learned visual semantics of the other samples in each training step. Specifically, we first select the samples that are excluded in $\mathcal{L}_{\text{diff-max}}$, which can be expressed as $S = \{n|n = 1, ..., N\} - \{n_{\max}\}$, where $n_{\max}$ is the index of the target sample in $\mathcal{L}_{\text{diff-max}}$ and $S$ is the index set of the selected samples. Then, we use the momentum denoising U-Net $\tilde{\epsilon}_\theta$ to apply regularization for those samples represented by $S$, with the masked preserving loss as

$$\mathcal{L}_{\text{pres}} = \mathbb{E}_{n \in S,k,t}\left[\left\|\tilde{\epsilon}_\theta(z^{(t)}_{nk}, t, e_n) \odot M'_{nk} - \epsilon_\theta(z^{(t)}_{nk}, t, e_n) \odot M'_{nk}\right\|^2_2\right], \tag{7}$$

where $\tilde{\epsilon}_\theta$ is updated from $\epsilon_\theta$ using EMA (Tarvainen & Valpola, 2017) with the smoothing coefficient $\beta = 0.99$, thereby sustaining the prior accumulated knowledge of $\epsilon_\theta$ in each training step. By encouraging the consistency between the output of $\epsilon_\theta$ and $\tilde{\epsilon}_\theta$ in $\mathcal{L}_{\text{pres}}$, we can facilitate the knowledge maintenance of the other samples while learning a specific sample in $\mathcal{L}_{\text{diff-max}}$. Finally, using a loss weight $\lambda_{\text{pres}} = 0.5$, the total loss of the DS-Bal stage is formulated as

$$\mathcal{L}_{\text{DS-Bal}} = \mathcal{L}_{\text{diff-max}} + \lambda_{\text{pres}}\mathcal{L}_{\text{pres}} + \lambda_{\text{attn}}\mathcal{L}_{\text{attn}}. \tag{8}$$

# 4 EXPERIMENTS

## 4.1 EXPERIMENTAL SETUP

**Dataset, Implementation, and Evaluation.** For a systemic investigation, we collect a dataset from various domains, including characters, animation, buildings, objects, and animals. We employ Stable Diffusion 2.1 (Rombach et al., 2022) as the pretrained T2I diffusion model. Reference images are resized to $512 \times 512$, and the LoRA rank and alpha are set to 32. For the warm-up and DS-Bal stage, we set the training steps to 200 and 300 and the learning rate to 1e-4 and 1e-5, using AdamW (Loshchilov & Hutter, 2017) as the optimizer. To generate evaluation images, we carefully design 20 text prompts covering extensive situations. For each method, we generate 14,720 images to conduct a comprehensive evaluation. To ensure fairness, all the seeds are fixed during training and inference.

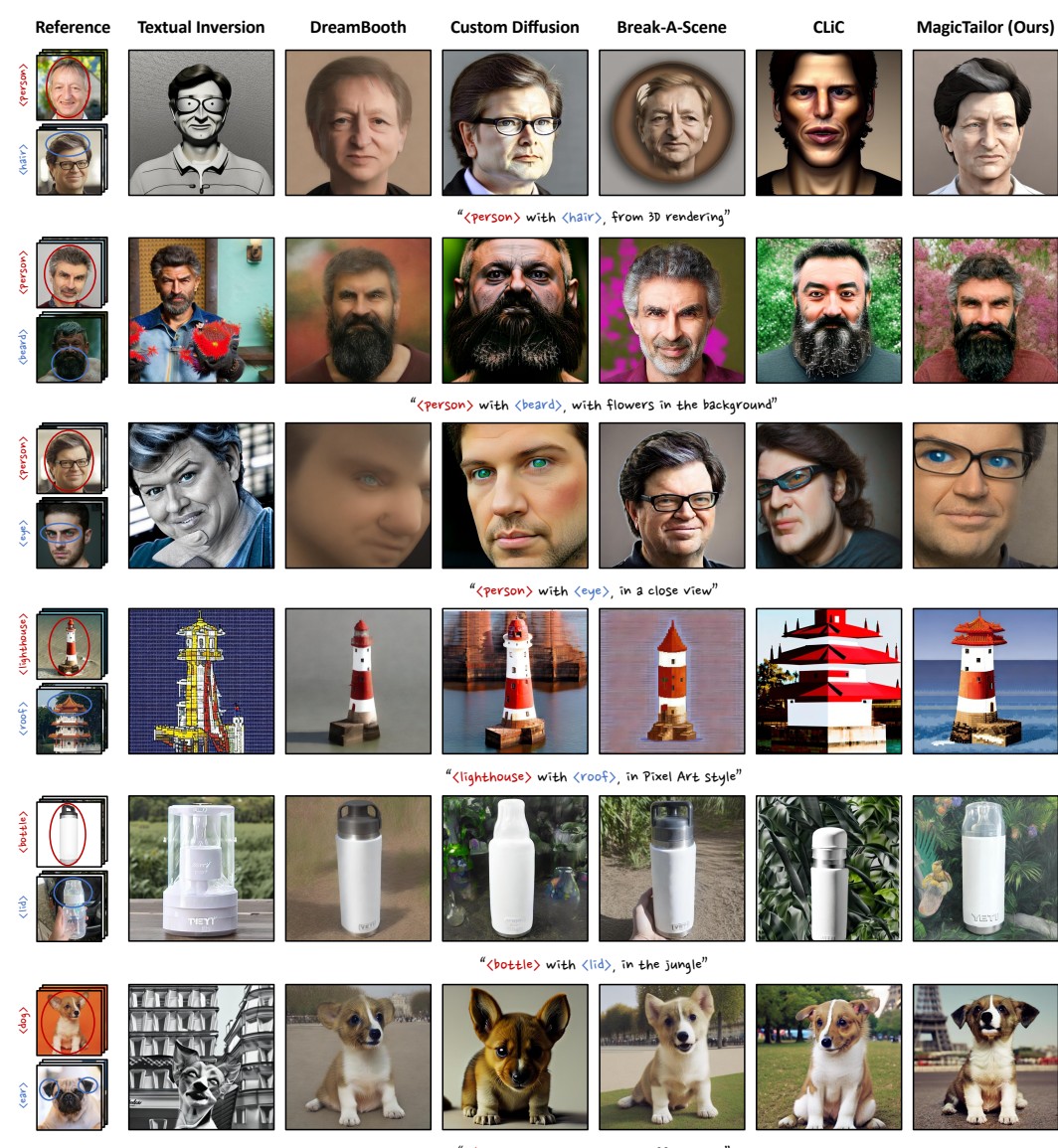

Figure 4: **Qualitative comparisons.** We present images generated by MagicTailor and the compared methods for various domains. MagicTailor generally achieves promising text alignment, strong identity fidelity, and high generation quality. More results are provided in Appendix D.

**Compared Methods.** We compare MagicTailor with SOTA methods of personalization, including Textual Inversion (TI) (Gal et al., 2022), DreamBooth (DB) (Ruiz et al., 2023), Custom Diffusion (CD) (Kumari et al., 2023), Break-A-Scene (BAS) (Avrahami et al., 2023), and CLiC (Safaee et al., 2024). For a fair and meaningful comparison, they are adapted to our task with minimal modification, *i.e.*, incorporating the masked diffusion loss (Equation 1) into them. Except for method-specific configurations, all methods use the same implementation above to ensure fairness. Due to the space limit, more details of the experimental setup are provided in Appendix A.

## 4.2 QUALITATIVE COMPARISONS

The qualitative results are presented in Figure 4. It shows that TI, CD, and CLiC mainly suffer from semantic pollution, where undesired visual semantics severely influence the personalized concept and even the other parts in generated images. Besides, we can observe that DB and BAS also underperform in this tough task. These methods exhibit an overemphasis on the concept or component due

Table 1: **Quantitative comparisons.** we compare our MagicTailor with SOTA methods of personalization based on automatic metrics and user study. The best results are marked in bold.

| Methods | Automatic Metrics | | | | User Study | | |
|---|---|---|---|---|---|---|---|
| | CLIP-T ↑ | CLIP-I ↑ | DINO ↑ | DreamSim ↓ | Text Align. ↑ | Id. Fidelity ↑ | Gen. Quality ↑ |
| Textual Inversion (Gal et al., 2022) | 0.236 | 0.742 | 0.620 | 0.558 | 5.8% | 2.5% | 5.2% |
| DreamBooth (Ruiz et al., 2023) | 0.266 | 0.841 | 0.798 | 0.323 | 15.3% | 14.7% | 12.5% |
| Custom Diffusion (Kumari et al., 2023) | 0.251 | 0.797 | 0.750 | 0.407 | 7.1% | 7.7% | 9.8% |
| Break-A-Scene (Avrahami et al., 2023) | 0.259 | 0.840 | 0.780 | 0.338 | 10.8% | 12.1% | 22.8% |
| CLiC (Safaee et al., 2024) | 0.263 | 0.764 | 0.663 | 0.499 | 4.5% | 5.1% | 6.2% |
| MagicTailor (Ours) | **0.270** | **0.854** | **0.813** | **0.279** | **56.5%** | **57.9%** | **43.4%** |

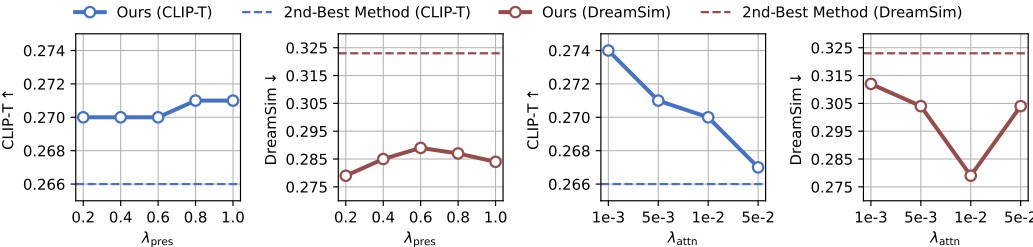

Figure 5: **Ablation of loss weights.** we report CLIP-T for text alignment, and DreamSim for identity fidelity as it is most similar to human judgments (Fu et al., 2023). For reference, we also present the results of the second-best method in Table 1, highlighting our robustness on loss weights.

to semantic imbalance, which even leads to an absence of the target component. Moreover, there is also an interesting observation that imbalanced learning could aggravate the effect of semantic pollution, making the color and texture of the target concept or component mistakenly transferred into an unexpected part of generated images. Compared with these methods, MagicTailor can achieve superior performance in generating text-aligned images that also faithfully reflect the target concept and component, demonstrating its remarkable performance in this newly formulated task.

## 4.3 QUANTITATIVE COMPARISONS

**Automatic Metrics.** We utilize four automatic metrics of the aspects of text alignment (CLIP-T (Gal et al., 2022)) and identity fidelity (CLIP-I (Radford et al., 2021), DINO (Oquab et al., 2023), DreamSim (Fu et al., 2023)). To precisely measure identity fidelity, we segment out the concept and component in each reference and evaluation image, and then eliminate the target component from the segmented concept (see the detailed setup in Appendix A). As we can see, component-controllable personalization remains a tough task even for SOTA methods of personalization. By comparison, MagicTailor can achieve the best identity fidelity and the second-best text alignment. It should be credited to the utilization of an effective framework tailored to this special task.

**User Study.** We further evaluate the methods with a user study. Specifically, a detailed questionnaire is designed to display 20 groups of evaluation images with the corresponding text prompt and reference images. Users are asked to select the best result in each group for three aspects, including text alignment, identity fidelity, and generation quality (see the detailed setup in Appendix A). Finally, we collect a total of 3,180 valid answers and report the selected rates in Table 1. It can be observed that MagicTailor can achieve outstanding performance in human preferences, showcasing its effectiveness in component-controllable personalization.

## 4.4 ABLATION STUDIES

We conduct comprehensive ablation studies of MagicTailor, aiming to verify the capability of the overall pipeline. For more ablation studies on other aspects, please refer to Appendix C.

Table 2: **Ablation of key techniques.** Our DM-Deg and DS-Bal effectively contribute to a superior performance trade-off.

| DM-Deg | DS-Bal | CLIP-T ↑ | CLIP-I ↑ | DINO ↑ | DreamSim ↓ |
|--------|--------|----------|----------|--------|------------|
|        |        | 0.275    | 0.837    | 0.798  | 0.317      |
| ✓      |        | **0.276** | 0.848   | 0.809  | 0.294      |
|        | ✓      | 0.270    | 0.845    | 0.802  | 0.304      |
| ✓      | ✓      | 0.270    | **0.854** | **0.813** | **0.279** |

Table 3: **Ablation of DS-Bal.** We compare DS-Bal with its variants, showing its excellence.

| U-Net Variants | CLIP-T ↑ | CLIP-I ↑ | DINO ↑ | DreamSim ↓ |
|----------------|----------|----------|--------|------------|
| Fixed ($\beta = 0$) | 0.268 | 0.850 | 0.803 | 0.293 |
| Fixed ($\beta = 1$) | 0.270 | 0.851 | 0.808 | 0.286 |
| Momentum ($\beta = 0.5$) | 0.268 | 0.850 | 0.805 | 0.290 |
| Momentum ($\beta = 0.9$) | 0.269 | 0.850 | 0.808 | 0.288 |
| Momentum (Ours) | **0.270** | **0.854** | **0.813** | **0.279** |

Table 4: **Ablation of DM-Deg.** We compare DM-Deg with its variants and the mask-out strategy. Our DM-Deg attains superior overall performance on text alignment and identity fidelity.

| Intensity Variants | CLIP-T ↑ | CLIP-I ↑ | DINO ↑ | DreamSim ↓ |
|--------------------|----------|----------|--------|------------|
| Mask-Out Startegy | 0.270 | 0.818 | 0.760 | 0.375 |
| Fixed ($\alpha = 0.4$) | 0.270 | 0.849 | 0.800 | 0.297 |
| Fixed ($\alpha = 0.6$) | 0.271 | 0.845 | 0.794 | 0.310 |
| Fixed ($\alpha = 0.8$) | 0.271 | 0.846 | 0.796 | 0.305 |
| Linear (Ascent) | 0.270 | 0.846 | 0.797 | 0.307 |
| Linear (Descent) | 0.261 | 0.851 | 0.802 | 0.300 |
| Dynamic ($\gamma = 8$) | 0.266 | 0.850 | 0.806 | 0.289 |
| Dynamic ($\gamma = 16$) | 0.268 | 0.854 | 0.813 | 0.282 |
| Dynamic ($\gamma = 64$) | **0.271** | 0.852 | 0.812 | 0.283 |
| Dynamic (Ours) | 0.270 | **0.854** | **0.813** | **0.279** |

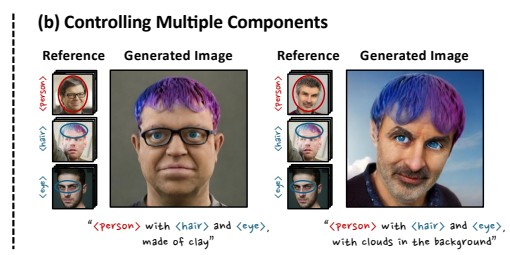

**(a) Decoupled Generation.** Reference — Generated Images
"⟨person⟩ with ⟨eye⟩, watercolor painting"  "⟨person⟩ dresses like batman"  "⟨eye⟩ in the cover of a wizard book"

**(b) Controlling Multiple Components.** Reference — Generated Image
"⟨person⟩ with ⟨hair⟩ and ⟨eye⟩, made of clay"  "⟨person⟩ with ⟨hair⟩ and ⟨eye⟩, with clouds in the background"

Figure 6: **(a) Decoupled Generation.** MagicTailor can also separately generate the target concept and component, enriching prospective combinations. **(b) Controlling multiple components.** MagicTailor shows the potential to handle more than one component, highlighting its effectiveness.

**Effectiveness of Key Techniques.** In Table 2, we investigate two key techniques of MagicTailor by starting from a baseline framework described in Section 3.1. Even without DM-Deg and DS-Bal, such a baseline framework can still have competitive performance, showing its reliability. On top of that, we introduce DM-Deg and DS-Bal, where the superior performance trade-off indicates the significance of these two key techniques.

**Dynamic Intensity Matters.** In Table 4, we explore DM-Deg by comparing it with 1) the mask-out strategy; 2) the fixed intensity; 3) the linear intensity ($\alpha$ goes from 1 to 0, or from 0 to 1); and 4) the dynamic intensity with different $\gamma$. First, the terrible performance of the mask-out strategy verifies that it is not a good solution for semantic pollution. Moreover, the dynamic intensity generally shows better results, and it can achieve better overall performance with a proper $\gamma$.

**Momentum Denoising U-Net as a Good Regularizer.** In Table 3, we study DS-Bal by modifying the U-Net for regularization as 1) the fixed U-Net with $\beta = 0$ (*i.e.*, the one just after warm-up); 2) the fixed U-Net with $\beta = 1$ (*i.e.*, the one from the last step); and 3) the momentum U-Net with other $\beta$. The results demonstrate that employing the U-Net with a high momentum rate can yield better regularization to tackle semantic imbalance, thus leading to excellent performance.

**Sensitivity Analysis of Loss Weights.** In Figure 5, we analyze the sensitivity of loss weights in Equation 8 (*i.e.*, $\lambda_{\text{pres}}$ and $\lambda_{\text{attn}}$), since loss weights are often critical for model training. As we can see, when $\lambda_{\text{pres}}$ and $\lambda_{\text{attn}}$ vary within a reasonable range, our MagicTailor can consistently attain SOTA performance, revealing the robustness of MagicTailor on loss weights.

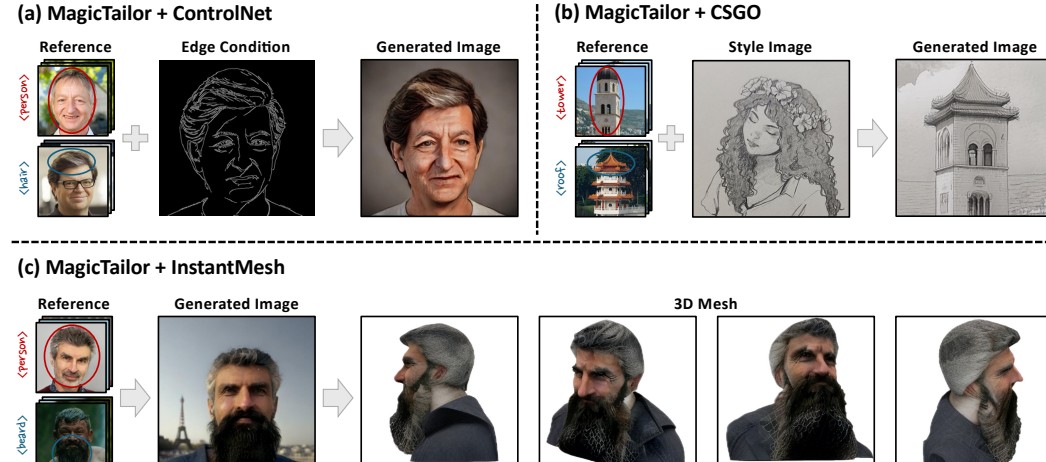

Figure 7: **Enhancing other generative tools.** MagicTailor can conveniently collaborate with a variety of generative tools that focus on other tasks, equipping them with an additional ability to control the concept's component in their pipelines.

### 4.5 FURTHER APPLICATIONS

**Decoupled Generation.** After learning from a concept-component pair, MagicTailor can also enable decoupled generation. As shown in Figure 6(a), MagicTailor can generate the target concept and component separately in various and even cross-domain contexts. This should be credited to its remarkable ability to accurately capture different-level visual semantics. Such an ability extends the flexibility of the possible combination between the concept and component.

**Controlling Multiple Components.** In this paper, we focus on personalizing one concept and one component, because such a setting is enough to cover extensive scenarios in the real world, and can be further extended to reconfigure multiple components with an iterative procedure. However, as shown in Figure 6(b), our MagicTailor also exhibits the potential to handle one concept and multiple components simultaneously. These results reflect a prospective direction of exploring better control over diverse components for a single concept.

**Enhancing Other Generative Tools.** In Figure 7, we provide some interesting method combinations to show that our MagicTailor can enhance other generative tools. The combined tools include ContorlNet (Zhang et al., 2023), CSGO (Xing et al., 2024), and InstantMesh (Xu et al., 2024). As we can see, MagicTailor can be seamlessly integrated into these tools, furnishing them with an additional ability to control the concept's component in their pipelines. For instance, working with MagicTailor, InstantMesh can conveniently achieve fine-grained design of 3D mesh, demonstrating the practicability of MagicTailor in collaborative applications.

## 5 CONCLUSION

In this paper, we introduce the novel task of *component-controllable personalization*, which allows for precise customization of individual components within a personalized concept. We tackle two major challenges that make this task particularly difficult: *semantic pollution*, where unwanted visual elements disrupt the integrity of the concept, and *semantic imbalance*, which skews the learning process of visual semantics. To address these challenges, we present *MagicTailor*, an innovative framework featuring *Dynamic Masked Degradation (DM-Deg)* to mitigate unwanted visual semantics and *Dual-Stream Balancing (DS-Bal)* to ensure balanced learning of visual components. Our comprehensive experiments demonstrate that MagicTailor not only sets a new benchmark in this challenging task but also opens up exciting possibilities for a wide range of creative applications. Looking ahead, we envision extending our approach to other areas of image and video generation, exploring how multi-level visual semantics can be recognized, controlled, and manipulated to unlock even more sophisticated and imaginative generative capabilities.

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

## A    More Details of Experimental Setup

### A.1    Dataset

As there is no existing dataset specifically for component-controllable personalization, we curate a dataset from the internet to conduct experiments. Unlike previous works (Ruiz et al., 2023; Kumari et al., 2023) that focus on very few categories of concepts, the dataset contains various domains of concepts and components, such as characters, animation, buildings, objects, and animals. Overall, the dataset consists of 23 concept-component pairs totally with 138 reference images, where each concept/component contains 3 reference images and a corresponding category label. The scale of this dataset is aligned with the scale of those datasets used in compared methods (Gal et al., 2022; Ruiz et al., 2023; Kumari et al., 2023; Avrahami et al., 2023; Safaee et al., 2024).

### A.2    Implementation

We utilize Stable Diffusion 2.1 (Rombach et al., 2022) as the pretrained T2I diffusion model. As commonly done, the resolution of reference images is set to $512 \times 512$. Besides, the rank and alpha of the LoRA module are set to 32. For the warm-up/DS-Bal stage, we set the learning rate to 1e-4/1e-5 and the training steps to 200/300. Moreover, the learning rate is further scaled by the batch size, which is set to completely contain a concept-component pair. MagicTailor is trained with an AdamW (Loshchilov & Hutter, 2017) optimizer and a DDPM Ho et al. (2020) sampler on an NVIDIA A100 GPU. For one concept-component pair, it runs for about 5 minutes. All experiments are accomplished with Python 3.10.11 and PyTorch 1.13.1, based on CUDA 11.6. Following (Avrahami et al., 2023), the tensor precision is set to float16 to accelerate training. For a fair comparison, all random seeds are fixed at 0 in each experiment, and all compared methods use the same implementation above except for method-specific configurations.

### A.3    Text Prompts For Evaluation

To generate images for evaluation, we carefully design 20 text prompts covering extensive situations, which are listed in Table 5. These text prompts can be divided into four aspects, including recontextualization, restylization, interaction, and property modification, where each aspect is composed of 5 text prompts. In recontextualization, we change the contexts to different locations and periods. In restylization, we transfer concepts into various artistic styles. In interaction, we explore the spatial interaction with other concepts. In property modification, we modify the properties of concepts in rendering, views, and materials. Such a group of diverse text prompts allows us to systemically evaluate the generalization capability of a method.

### A.4    Scheme of Generating Evaluation Images

We generate 32 images per text prompt for each pair, using a DDIM (Song et al., 2020) sampler with 50 steps and a classifier-free guidance scale of 7.5. To ensure fairness, we fix the random seed within the range of [0, 31] across all methods. This process results in a total of 14,720 images for each method to be evaluated, ensuring a robust and thorough comparison.

### A.5    Automatic Metrics

We utilize four automatic metrics in the aspects of text alignment (CLIP-T (Gal et al., 2022)) and identity fidelity (CLIP-I (Radford et al., 2021), DINO (Oquab et al., 2023), DreamSim (Fu et al., 2023)). To precisely measure identity fidelity, we improve the traditional measurement approach for personalization. This is because a reference image of the target concept/component could contain an undesired component/concept that is not expected to appear in evaluation images. Specifically, we use Grounded-SAM (Ren et al., 2024) to segment out the concept and component in each reference and evaluation image. Then, we further eliminate the target component from the segmented concept, *e.g.*, eliminate the hair from the person in a "<person> + <hair>" pair. Such a process is similar to the one adopted in (Avrahami et al., 2023). As a result, using the segmented version of evaluation images and reference images, we can accurately calculate the metrics of identity fidelity.

Table 5: **Text prompts used to generate evaluation images.** These text prompts can be divided into four aspects: recontextualization, restylization, interaction, and property modification, covering extensive situations to systemically evaluate the generalization capability of a method. Note that "<placeholder>" will be replaced by the combination of pseudo-words (*e.g.*, "<tower> with <roof>") when generating evaluation images, and will be replaced by the combination of category labels (*e.g.*, "tower with roof") when calculating the metric of text alignment.

| Recontextualization | Restylization |
| --- | --- |
| "<placeholder>, on the beach" | "<placeholder>, watercolor painting" |
| "<placeholder>, in the jungle" | "<placeholder>, Ukiyo-e painting" |
| "<placeholder>, in the snow" | "<placeholder>, in Pixel Art style" |
| "<placeholder>, at night" | "<placeholder>, in Von Gogh style" |
| "<placeholder>, in autumn" | "<placeholder>, in a comic book" |
| Interaction | Property Modification |
| "<placeholder>, with clouds in the background" | "<placeholder>, from 3D rendering" |
| "<placeholder>, with flowers in the background" | "<placeholder>, in a far view" |
| "<placeholder>, near the Eiffel Tower" | "<placeholder>, in a close view" |
| "<placeholder>, on top of water" | "<placeholder>, made of clay" |
| "<placeholder>, in front of the Mount Fuji" | "<placeholder>, made of plastic" |

## A.6 USER STUDY

In addition to using automatic metrics, we further evaluate the methods with a user study. Specifically, we design a questionnaire to display 20 groups of evaluation images generated by our method and other methods. Besides, each group also contains the corresponding text prompt and the reference images of the concept and component, where we adopt the same text prompts that are used to calculate CLIP-T. The results of our method and all the compared methods are presented on the same page. Clear rules are established for users to evaluate in three aspects, including text alignment, identity fidelity, and generation quality. Users are requested to select the best result in each group by answering the corresponding questions of these three aspects. We hide all the method names and randomize the order of methods to ensure fairness. Finally, 3,180 valid answers are collected for a sufficient quantitative evaluation of human preferences.

## A.7 COMPARED METHODS

In our experiments, we compare MagicTailor with SOTA methods in the domain of personalization, including Textual Inversion (TI) (Gal et al., 2022), DreamBooth-LoRA (DB) (Ruiz et al., 2023), Custom Diffusion (CD) (Kumari et al., 2023), Break-A-Scene (BAS) (Avrahami et al., 2023), and CLiC (Safaee et al., 2024). We adopt the LoRA version of DreamBooth because it generally shows better generation performance. For TI, DB, and CD, we use the third-party implementation in Diffusers [1]. For BAS, we use the official implementation. For CLiC, we reproduce it following the resource paper as the official code is not released. Unless otherwise specified, method-specific configurations are set up by following their resource papers or Diffusers. We empirically adjust the learning rate of CD and CLiC to 1e-4 and 5e-5 respectively, because they perform very poorly with the original learning rates. For a fair and meaningful comparison, these methods should be adapted to our task setting with minimal modification. Therefore, we integrate the masked diffusion loss (Equation 1) into them while using the same segmentation masks from MagicTailor.

---

[1]https://huggingface.co/docs/diffusers/index

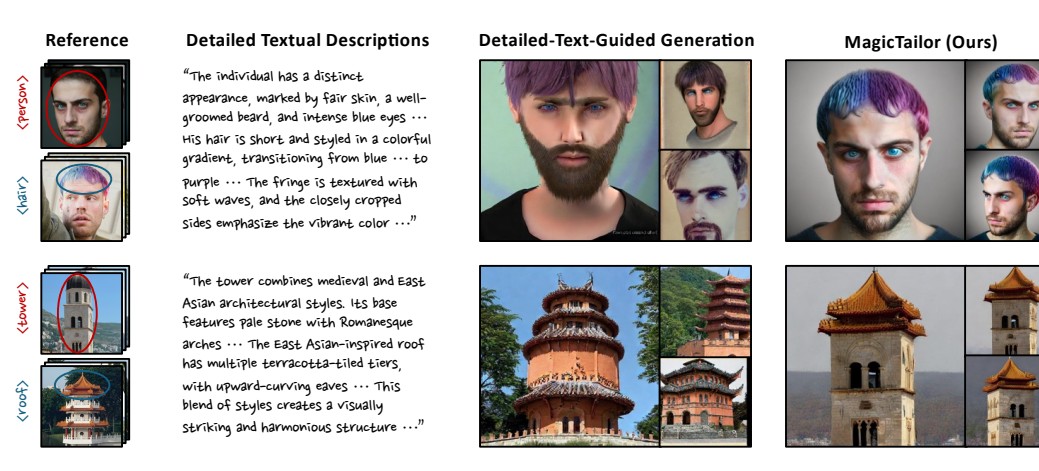

Figure 8: **Comparing with detailed-text-guided generation.** We use GPT-4o to generate and merge detailed textual descriptions for the target concept and component, which are fed into Stable Diffusion 2.1 to conduct text-to-image generation. This paradigm cannot perform well and produce inconsistent images, while MagicTailor can achieve faithful and consistent generation.

## B    ADDITIONAL COMPARISONS

### B.1    COMPARING WITH DETAILED-TEXT-GUIDED GENERATION

One might be curious about whether component-controllable personalization can be accomplished by providing detailed textual descriptions to the T2I model. To investigate this, we separately feed the reference images of the concept and component into GPT-4o[2] to obtain detailed textual descriptions for them. The text prompt we used is "Please detailedly describe the <concept/component> of the upload images in a parapraph", where "<concept/component>" is replaced with the category label of the concept or component. Then, we ask GPT-4o to merge these textual descriptions using natural language, and input them into the Stable Diffusion 2.1 (Rombach et al., 2022) to generate the corresponding images. Some examples for a qualitative comparison are shown in Figure 8. As we can see, such an approach cannot achieve satisfactory results, because it is hard to guarantee that visual semantics can be completely expressed by using the combination of text tokens. In contrast, our MagicTailor is able to accurately learn the desired visual semantics of the concept and component from reference images, and thus lead to consistent and excellent generation in this tough task.

### B.2    COMPARING WITH COMMERCIAL MODELS

It is also worth exploring whether existing commercial models, which can understand and somehow generate both text and images by themselves or other integrated tools, are capable of handling component-controllable personalization. We choose two widely recognized commercial models, GPT-4o and Gemini 1.5 Flash[3], for a qualitative comparison. First, we separately feed the reference images of the concept and component into them, along with the text prompt of "The uploaded images contain a special instance of the <concept/component>, please mark it as #<concept/component>", where "<concept/component>" is replaced with the category label of the concept or component. Then, we instruct them to perform image generation, using the text prompt of "Please generate images containing #<concept> with #<component>", where "<concept>" and "<component>" are replaced with the category label of the concept and component, respectively. As shown in Figure 9, these models struggle to reproduce the given concept, let alone reconfigure the concept's component. Whereas, our MagicTailor achieves superior results in component-controllable personalization, using a dedicated framework designed for this task. It demonstrates that, even though large commercial models are able to tackle multiple general tasks, there is also plenty of room for the community to explore specialized tasks for real-world applications.

---

[2]https://platform.openai.com/docs/models/gpt-4o
[3]https://gemini.google.com/

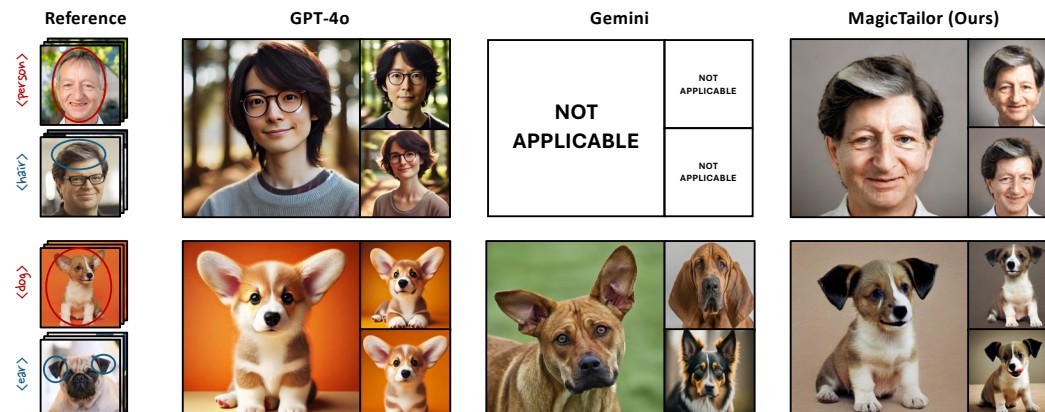

Figure 9: **Comparing with commercial models.** We input the reference images of the target concept and component to GPT-4o and Gemini, along with structured text prompts, for conducting image generation. Even though capable of handling multiple general tasks, these models still fall short in this task. In contrast, our MagicTailor performs well using a dedicated framework.

Table 6: **Ablations of warm-up.** We compare MagicTailor with the variant that removes warm-up. The results exhibit the significance of the warm-up stage for the framework of MagicTailor.

| Warm-up Variants | CLIP-T ↑ | CLIP-I ↑ | DINO ↑ | DreamSim ↓ |
|---|---|---|---|---|
| w/o Warm-up | **0.272** | 0.844 | 0.793 | 0.320 |
| w/ Warm-up (Ours) | 0.270 | **0.854** | **0.813** | **0.279** |

## C    ADDITIONAL ABLATION STUDIES

### C.1    NECESSITY OF WARM-UP TRAINING

In MagicTailor, we start with a warm-up phase for the T2I model to preliminarily inject the knowledge for the subsequent phase of DS-Bal. Here we investigate the necessity of such a warm-up phase for generation performance. In Table 6, when removing the warm-up phase, even though Magic-Tailor could obtain slight improvement in text alignment, it severely suffers from the huge drop in identity fidelity. This is because such a scheme makes it difficult to construct a decent momentum denoising U-Net for DS-Bal. Whereas integrated with a warm-up phase, MagicTailor can achieve superior overall performance, which is attributed to the knowledge reserved from warm-up.

### C.2    PERFORMANCE ON DIFFERENT NUMBERS OF REFERENCE IMAGES

As described in Appendix A, each concept/component contains 3 reference images in a concept-component pair of the dataset. Here we change the number of reference images to analyze the performance variation of MagicTailor. The qualitative results are presented in Figure 10. When the number of reference images is reduced, MagicTailor can still show satisfactory performance. This also demonstrates that, while more reference images could lead to better generalization ability, one reference image per concept/component is enough to obtain a decent result with our MagicTailor.

### C.3    ROBUSTNESS ON LINKING WORDS

Generally, we use "with" to link the pseudo-words of the concept and component in a text prompt, *e.g.*, "<person> with <beard>, in Von Gogh style". Here we evaluate the robustness of our method on different linking words. We choose several words, which are commonly used to indicate ownership or association, to construct text prompts and then feed them into the same fine-tuned T2I model.

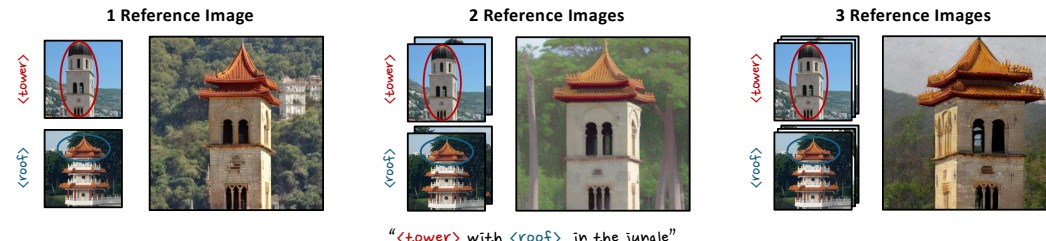

Figure 10: **Ablation of the number of reference images.** We present qualitative results to show that MagicTailor can still achieve satisfactory performance when provided only 1 or 2 reference images per concept and component.

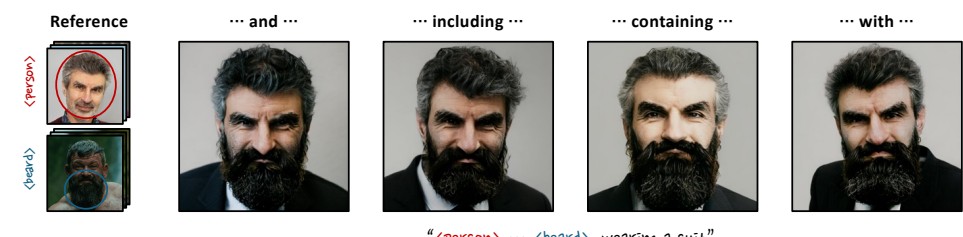

Figure 11: **Ablation of the linking word.** We present qualitative results generated with different linking words, showing the robustness of MagicTailor.

As shown in Figure 11, the generation performance of our MagicTailor remains robust regardless of the linking word used, exhibiting its flexibility to textual descriptions.

## D   MORE QUALITATIVE RESULTS

In Figure 12, we provide more evaluation images for a substantial qualitative comparison. It can be clearly observed that semantic pollution remains an intractable problem for these compared methods. While employing the masked diffusion loss, they still fall short in suppressing the appearance of undesired visual semantics. This is due to the leak of an effective mechanism to alleviate the T2I model's perception for these semantics. To address this, our MagicTailor utilizes Dynamic Masked Degradation (DM-Deg) to dynamically perturb undesired visual semantics during the learning phase, and thus achieve better performance. On the other hand, the compared methods are also severely influenced by semantic imbalance, resulting in overemphasis or even overfitting on the concept or component. This is because the inherent imbalance of visual semantics complicates the learning process. In response to this issue, our MagicTailor applies Dual-Stream Balancing (DS-Bal) to balance the learning of visual semantics, effectively showcasing its prowess in this tough task.

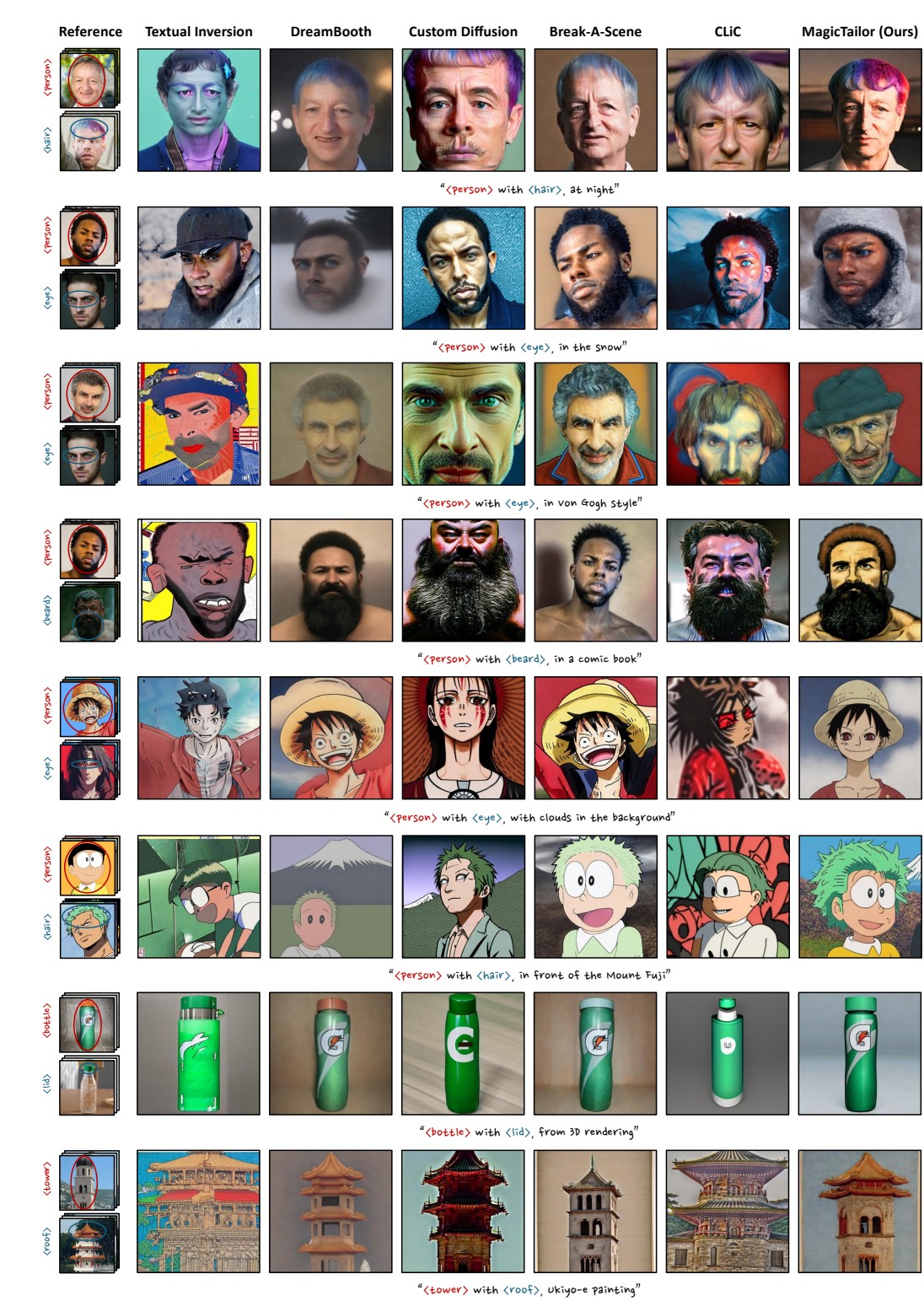

Figure 12: **More qualitative comparisons.** We present images generated by our MagicTailor and SOTA methods of personalization for various domains including characters, animation, buildings, objects, and animals. MagicTailor generally achieves promising text alignment, strong identity fidelity, and high generation quality.

