# OpenReview forum: "MagicTailor: Component-Controllable Personalization in Text-to-Image Diffusion Models"
_ICLR.cc/2025/Conference — ICLR 2025 Conference Withdrawn Submission_

### Official Review · Reviewer_yGmY · 2024-10-30

**Soundness:** 2
**Presentation:** 3
**Contribution:** 2
**Rating:** 3
**Confidence:** 4

**Summary:**

The authors of the paper propose a new way of customization of text-to-image synthesis. Instead of customizing by concepts, they propose to customize the components of concepts and synthesis new images with a combination of components and concepts. They also highlight 2 main problems they meet in customization process and propose solutions for them.

**Strengths:**

1. The authors composed a diverse quantitative and qualitative comparisons. They addressed the problem holistically by created ablation studies, user studies, sensitivity studies for hyper parameters, etc.
2. The idea of customizing components and combine them together in a new generated image is relatively new.

**Weaknesses:**

1. The semantic pollution and semantic imbalance issues are too similar and not clearly disentangled. I am not convinced why the authors addressed them as different problems.
2. Lack of evidence to claim the 2 aforementioned problems occur. Only a couple of examples are shown in support.
3. The Dynamic Masked Degradation and Dual-Stream Balancing solutions also seem very similar. They use similar masking approaches. The separate approach of Sample-wise Min-Max Optimization and Selective Preserving Regularization is not clearly justified and seem overcomplication.
4. Lack of novelty in proposed solutions. The masking and pixel degradations methods are not new. They often are used to enhance diversity, avoid overfitting, and localize contributions in text-to-image generation.
5. Not fair comparisons. The authors only used competitors (Text Inversion, DreamBooth, Custom Diffusion, etc..) which are baseline methods in customization and personalization tasks. There are many new methods (such as [1,2,3,4,5, ...]) with new state of the art results
6. Bad quality of results. The generated images in comparisons have poor quality and poor fidelity.
7. In L259-L260, the authors claim that the model may memorize the introduced noise, and the memorization strength is correlated with the training steps. However, in the ablation study in Tab 4, the Linear (Ascent) and Linear (Descent), which are ascending and descending w.r.t training steps, have almost similar results in all metrics. This evidence works agains their claim.



[1] "Customization Assistant for Text-to-image Generation" Yufan Zhou, et al.

[2] "Orthogonal Adaptation for Modular Customization of Diffusion Models" Ryan Po, et al.

[3] "Attention Calibration for Disentangled Text-to-Image Personalization" Yanbing Zhang, et al.

[4] "JeDi: Joint-Image Diffusion Models for Finetuning-Free Personalized Text-to-Image Generation" Yu Zeng, et al.

[5] "A Neural Space-Time Representation for Text-to-Image Personalization" Yuval Alaluf, et al.

**Questions:**

Address to the weaknesses mentioned above, please.

---

### Official Review · Reviewer_43Zx · 2024-11-03

**Soundness:** 3
**Presentation:** 3
**Contribution:** 2
**Rating:** 5
**Confidence:** 4

**Summary:**

This paper presents MagicTailor, which achieves fine-grained concept customization for text-to-image diffusion models. Novel task component-controllable personalization is introduced, which allows user to reconfigure and personalize specific components of concepts. Different from existing personalization methods, MagicTailor enable model to capture local appearance from reference images, and generative more creative images with mixture of fine-grained concepts. The paper primarily addresses two challenging issues in component-controllable personalization: semantic pollution and semantic imbalance. The authors clearly explained the above two critical effects in the manuscript, and accordingly present different approaches to alleviate the semantic issues. The authors compare with several existing personalization baselines both quantitatively and qualitatively, and demonstrates the effectiveness of proposed MagicTailor.

**Strengths:**

1. The writing of the paper is clear and easy to follow.
2. This work defines a task named component-controllable personalization, which aims at more precise and fine-grained customization for T2I models,  and this may be of interest to the community in a broader context.
3. The overall structure of the article is clear. Two challenges of component-controllable personalization are discussed, and visual effects are illustrated for better comprehension. Further, DM-Deg and DS-Bal are proposed respectively to handle the above semantic issues.
4. The visual results presented in this paper provide compelling evidence that MagicTailor method surpasses current competing techniques in fine-grained component-controllbale generation for diffusion models.
5. This work contributes a collection of paired data for the proposed novel task, which makes the work easier to be followed by future works.

**Weaknesses:**

1. The designed method is a little bit complicated. It seems that undesired conflicts may occur during the learning processes of different concepts. Hence the authors present multiple carefully designed constraints for different 'unfortunate' occasions. I wonder whether the proposed methods generalize well when applied to different model architectures, will the hyper-params be carefully chosen again? As a personalization method, is it compatible with some trained LoRAs (e.g. cartoon, realistic) from the community? More experiments should be conducted to test the compatibility of MagicTailor with existing pretrained LoRAs and other model architectures. This would help address the concerns about generalizability and provide valuable information for potential users of the method.
2. Visual results of multiple concepts concepts are limited, I don't find comprehensive discussion when the number of reference fine-grained concepts increase to 4,5 or even more. Will the model suffer from fusion problems under such circumstance? More visual results should be provided (in Fig. 6 (b)) to test the boundary of MagicTailor on multiple concepts generation.
3. It seems that MagicTailor mainly combines concepts from areas which share same semantics (e.g., face from A and eyes from B), I wonder how MagicTailor performs in some more complicated scenarios, for example, I want to generate a beautiful girl wearing a T-shirt from person A, and she has eyes from boy B, and is holding a small dog shown in Picture C. I suggest the authors to study some more complex cases, where different semantic parts from different objects are specified by users, and provide some visual results or quantitative analysis. This would provide valuable insight into the method's limitations and potential applications.
4. Missing important related work, PartCraft [1], which also allows users to select semantic parts for creative personalization. With PartCraft as the pioneer work, The novelty of the task proposed in this paper needs to be rejudged. The authors should present a detailed discussion about the difference between MagicTailor and PartCraft to clearify their contribution. There are also many visual results in PartCraft by composing more than 3 parts from different reference images, the authors is supposed to present more visual comparisons in Figure. 6, so as to show how MagicTailor performs when the number of semantic parts increases. There are many meaningful visual cases from PartCraft (Fig. 7), the authors can refer to them and test some similar cases for more intuitive comprehension.

[1] Ng, Kam Woh, et al. "PartCraft: Crafting Creative Objects by Parts." European Conference on Computer Vision. Springer, Cham, 2025.

**Questions:**

Please refer to the weakness.

---

### Official Review · Reviewer_maLm · 2024-11-03

**Soundness:** 2
**Presentation:** 2
**Contribution:** 3
**Rating:** 5
**Confidence:** 4

**Summary:**

This paper proposes a new architecture for component-controllable personalization. It introduces two main design choices: Dynamic Masked Degradation (DM-Deg) to prevent semantic pollution by introducing noise to masked-out image regions, and Dual-Stream Balancing (DS-Bal) to avoid semantic imbalance through Sample-wise Min-Max Optimization and Selective Preserving Regularization.

**Strengths:**

**1. Task Definition:** Unlike prior works that usually focus on multi-concept personalization, this paper uniquely targets component-controllable personalization, which separates the concept and the component and allows control over specific components (e.g., a person's eyebrow) within a larger concept (e.g., the person).

**2. Problem Statement:** The authors clearly depict the challenges they aim to solve (semantic pollution and imbalance), contributing to the community by highlighting a new problem to address.

**3. Comprehensive Evaluation:** The paper includes thorough quantitative and ablation results, showcasing the effectiveness of the proposed method.

**Weaknesses:**

**1. Complexity of Implementation:** The proposed methods, DS-Bal and DM-Deg, seem complex to implement and computationally demanding, requiring substantial resources.

**2. Limited Generalizability:** The paper primarily shows results in controlled or geometrically similar scenarios (e.g., where the layout in concept and composition are consistent), it raises questions about how the method would perform under more complex or varied conditions.

**Questions:**

**1. Generalizability:**

Is it possible to use concept/composition pairs that exhibit large geometric deformations?
For example, <person> from upper body portrait images and <eyebrow> from face images.
Another example could be generating an image such as "A photo of <dog> with <human ears>."
Can the model generate more dynamic images such as "A photo of <person> with <beard> riding a bicycle in front of the Eiffel Tower"?

**2. The effectiveness of DM-Deg:** Is DM-Deg more effective than using detailed prompts during training [1]? For instance, if detailed captions describing the image are provided (e.g., "A photo of a <sks> dog in a jungle background..."), the model could use textual prior knowledge to determine where to attend and effectively learn the <sks>.

**3. Complexity Analysis:** It would be helpful to provide an analysis of the time and memory complexity compared to other methods.

**4. SDXL:** The image quality seems limited because of the diffusion backbone. Is the model compatible with other better backbone?


[1] Kim, J., Park, J., & Rhee, W. (2024). Selectively Informative Description Can Reduce Undesired Embedding Entanglements in Text-to-Image Personalization. In Proceedings of the IEEE/CVF Conference on Computer Vision and Pattern Recognition (pp. 8312-8322).

---

### Note · Authors · 2024-11-14

I have read and agree with the venue's withdrawal policy on behalf of myself and my co-authors.